# HiRA: Parameter-Efficient Hadamard High-Rank Adaptation for Large Language Models

**Qiushi Huang**[1,2*], **Tom Ko**[3], **Zhan Zhuang**[1,4], **Lilian Tang**[2], **Yu Zhang**[1†]
[1]Southern University of Science and Technology, [2]University of Surrey
[3]ByteDance, [4]City University of Hong Kong
{qiushi.huang.cs, tomkocse, yu.zhang.ust}@gmail.com,
h.tang@surrey.ac.uk,12250063@mail.sustech.edu.cn

## Abstract

We propose Hadamard High-Rank Adaptation (HiRA), a parameter-efficient fine-tuning (PEFT) method that enhances the adaptability of Large Language Models (LLMs). While Low-rank Adaptation (LoRA) is widely used to reduce resource demands, its low-rank updates may limit its expressiveness for new tasks. HiRA addresses this by using a Hadamard product to retain high-rank update parameters, improving the model capacity. Empirically, HiRA outperforms LoRA and its variants on several tasks, with extensive ablation studies validating its effectiveness. Our code is available at `https://github.com/hqsiswiliam/hira`.

## 1 Introduction

Recent advancements in pre-trained Large Language Models (LLMs) (Touvron et al., 2023; Zhang et al., 2022; Achiam et al., 2023) have significantly enhanced performance across various natural language processing tasks. Traditionally, adapting those LLMs to specific tasks required full fine-tuning, wherein all model parameters are updated. However, due to the massive number of parameters in those LLMs, full fine-tuning becomes computationally prohibitive, especially in resource-constrained environments.

To address this challenge, parameter-efficient fine-tuning (PEFT) methods have been developed to adapt LLMs by updating only a small subset of parameters. Building on this approach, several recent studies (Lester et al., 2021; Liu et al., 2022; Hu et al., 2021; Liu et al., 2024) have introduced methods that maintain the integrity of the original architecture by freezing the majority of the model parameters and introducing updates to a limited set. Notably, LoRA (Hu et al., 2021) exemplifies PEFT by integrating a low-rank matrix decomposition into the update $\Delta W = L_1 L_2$, where $L_1 \in \mathbb{R}^{d \times r}$ and $L_2 \in \mathbb{R}^{r \times k}$ are low-rank matrices with the rank at most $r$. This technique significantly reduces computational costs required compared to updating the full-rank parameter matrix $W$.

However, previous studies (Jiang et al., 2024; Liu et al., 2023; 2024; Zhuang et al., 2024) have shown that LoRA and most of its variants (Lialin et al., 2023; Hayou et al., 2024) do not perform well when applied to complex tasks, such as commonsense reasoning that requires training on a single dataset but evaluating across multiple sub-tasks. One potential reason for LoRA's limitations in these scenarios is that its update matrix, $\Delta W$, which is derived from the multiplication of low-rank matrices $L_1$ and $L_2$, is confined to a maximum rank of $r$. Consequently, although $\Delta W$ is a $d \times k$ matrix, its rank cannot exceed $r$, which may limit the expressiveness of $\Delta W$, particularly for more complex tasks. A natural solution to this issue is to raise the rank of the update parameter matrix to increase its capability. However, due to resource constraints, we still hope to follow the PEFT strategy. This gives rise to our research question: "*Is it possible to achieve a higher-rank adaptation for LLMs under the PEFT strategy?*"

To answer this question, in this paper, we propose a Hadamard high-Rank Adaptation (HiRA) for LLMs. The central innovation of HiRA is to express the update parameter matrix $\Delta W$ as the Hadamard product (a.k.a. elementwise product) of the original parameter matrix in the LLM and a

---

*Work done as an intern at ByteDance.
†Corresponding author.

low-rank matrix to achieve a high-rank adaptation, thus increasing its rank and also expressiveness. Due to a property of the Hadamard product that $\text{Rank}(O_1 \odot O_2) \leq \text{Rank}(O_1) \times \text{Rank}(O_2)$ for two matrices $O_1$ and $O_2$ with an equal size (Million, 2007), where $\odot$ denotes the Hadamard product and $\text{Rank}(\cdot)$ gives the rank of a matrix, the Hadamard product could have a higher rank even though one of the two matrices in the Hadamard product is low-rank.

As an illustration, where the experimental setting is detailed in Appendix A.1, Figure 1 shows that the average rank of update parameters in HiRA is much higher than that in LoRA (i.e., 2837 vs. 32), which demonstrates that the proposed HiRA could possess high-rank update parameter matrices while keeping the same number of trainable parameters as LoRA. The increased rank of $\Delta W$ in HiRA could enhance its expressiveness, as demonstrated in our experimental section.

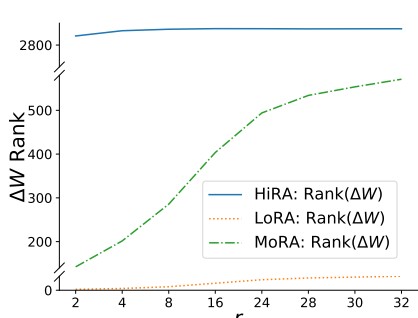

Figure 1: Rank comparison of update parameters $\Delta W$ among LoRA, MoRA, and the proposed HiRA.

Through comprehensive experiments, we demonstrate that HiRA significantly outperforms LoRA and its variants, showcasing the effectiveness of the proposed HiRA. Extensive ablation studies further elucidate the impact of different components in the HiRA, confirming its advantages and practical utility.

In summary, the contributions of this paper are as follows.

- We are the first to apply the Hadamard product to the parameter-efficient adaptation of LLMs.

- Based on the Hadamard product, we propose HiRA, a novel high-rank adaptation strategy for LLMs. The proposed HiRA method significantly increases the rank of update parameter matrices, enhancing the model's expressive power and adaptability.

- We conduct extensive experiments and analyses to demonstrate the effectiveness of HiRA.

## 2 RELATED WORKS

**Low-rank adaptation (LoRA).** PEFT methods aim to efficiently adapt LLMs to downstream tasks by training a small subset of additional parameters while keeping most pretrained weights frozen. Current prevalent PEFT methods can be broadly categorized into three main types, with low-rank adaptation being a major category. LoRA (Hu et al., 2021) factorizes the update weight into two low-rank matrices, allowing seamless integration into the original model without modifying its architecture or increasing inference overhead. As an extension, DoRA (Liu et al., 2024) decomposes weight updates into magnitude and direction, updating the directional component in the same manner as LoRA. MoRA (Jiang et al., 2024) compresses inputs via predefined functions, transforms them via a square "higher-rank" matrix, and then decompresses them to achieve a higher-rank adaptation. Different from LoRA which relies on the low-rank structure of the update weight, HiRA aims to learn the update weight with a high rank to enhance the expressive power. However, unlike MoRA, which relies on complex static compression and decompression functions that can complicate weight merging, HiRA merges seamlessly into the pretrained model, just as LoRA does.

Recently, FLoRA (Wen & Chaudhuri, 2024) uses the Hadamard product to implement an example-specific adapter, enabling efficient batching, while PACE (Ni et al., 2024) applies element-wise multiplicative noise to improve generalization cost-efficiently. In contrast, HiRA employs the Hadamard product to achieve a higher-rank adaptation (as empirically demonstrated in Figures 1 and 7) with a more in-depth analysis, which makes them significantly different.

**Other PEFT methods.** The second category comprises prompt-based methods, which integrate extra trainable virtual tokens into the input of LLMs and focus exclusively on training those tokens. Representative methods include Prompt Tuning (Lester et al., 2021), which introduces a series of virtual tokens for task-specific adaptations at the initial layer, and P-Tuning (Liu et al., 2022), which adds virtual tokens at every layer instead of the initial layer. Although prompt-based methods add a negligible number of trainable parameters into the input, they are notably sensitive to initialization

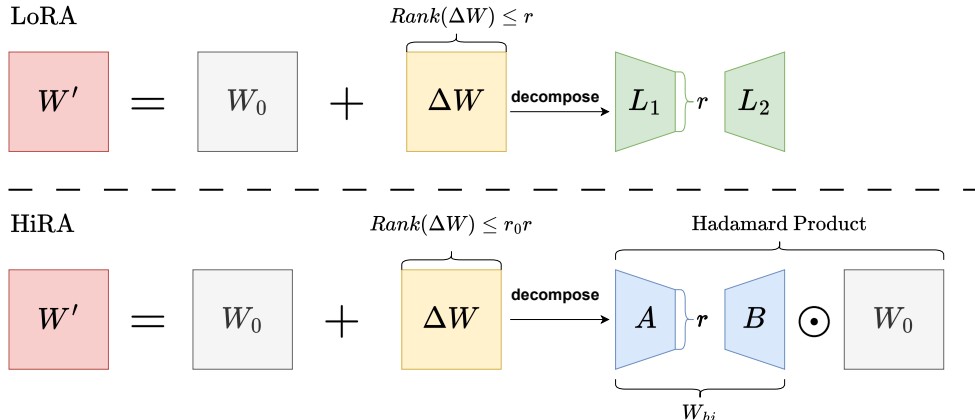

Figure 3: An illustration of the proposed HiRA method in comparison to the LoRA method.

(Wu et al., 2024). Moreover, due to the quadratic computational complexity of transformer architectures (Vaswani et al., 2017), prompt-based approaches could increase computational costs during inference proportionally to the length of the prompt. The third category includes adapter-based methods. Those methods insert trainable modules, such as adapter layers, into the original frozen LLMs. Typical methods include Adapters (Houlsby et al., 2019) and Compacter (Karimi Mahabadi et al., 2021) that add linear layers to LLMs. Parallel adapters (He et al., 2021) integrate adapters in parallel for performance enhancement. Those methods modify the architecture of original models during training and inference, potentially increasing overhead compared to the original LLMs.

## 3 MOTIVATIONS

Effective fine-tuning of LLMs requires a careful balance between model expressiveness and computational efficiency. Hence, existing studies use LoRA with lower ranks (e.g., 32 or 64). Such setting achieves good performance on some tasks. For more complex tasks such as commonsense reasoning, we find that LoRA with higher ranks can significantly enhance performance of Llama-3-8B as illustrated in Figure 2. Our results indicate that especially for complex tasks, updates on model parameters with a higher rank could be helpful to achieve better performance. This observation complements existing research by highlighting scenarios where high-rank adaptations offer significant advantages.

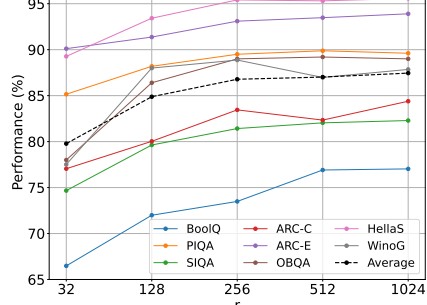

Figure 2: Performance of Llama-3-8B on commonsense reasoning using different LoRA configurations.

However, increasing the rank in LoRA results in heightened computational demands, which conflicts with the objectives of PEFT, and it becomes more difficult to train, often leading to issues like gradient explosion, as detailed in Appendix A.3. Therefore, there is a need for a method that enables high-rank adaptation without incurring additional computational burdens. The proposed HiRA method addresses this need, as introduced in the next section.

## 4 METHODOLOGY

### 4.1 RANK ANALYSIS

A limitation of LoRA and its variants relying on the product between two low-rank matrices is that the maximum achievable rank of the update parameter is inherently constrained. $W_0 \in \mathbb{R}^{d \times k}$ denotes the original parameter matrix and its rank is denoted by $r_0$, where $r_0 \leq \min(d, k)$. The update parameter matrix $\Delta W$ in LoRA is assumed to be the product of $L_1 \in \mathbb{R}^{d \times r}$ and $L_2 \in \mathbb{R}^{r \times k}$, where $r$ is much smaller than $d$ and $k$, i.e., $\Delta W = L_1 L_2$. Due to the property of the rank, the

maximum rank of $\Delta W$ is $r$. Thus, the low-rank property of $\Delta W$ may limit its capability to capture high-rank updates. As a result, such a low-rank update parameter may limit the rank of the final tuned parameter denoted by $W'$ (i.e., $W' = W_0 + \Delta W$) since

$$\text{Rank}(W_0 + \Delta W) \le \text{Rank}(W_0) + \text{Rank}(\Delta W) \le r_0 + r,$$

where the first equality holds due to the property of the rank function. Therefore, $W'$ has a maximum rank of $\min(\min(d, k), r_0 + r)$. Consequently, the low-rank property of $\Delta W$ may limit the expressiveness of $W'$. To address this, we propose the HiRA method to learn $\Delta W$ with a higher rank under the PEFT strategy, which could enhance expressiveness and performance.

## 4.2 ENHANCING THE RANK VIA HADAMARD PRODUCT

The Hadamard product of two matrices $P$ and $Q$ with the same size gives a matrix $O$ satisfying $o_{ij} = p_{ij}q_{ij}$, where $p_{ij}$, $q_{ij}$, and $o_{ij}$ denote the $(i, j)$th entry in $P$, $Q$, and $O$, respectively. So the Hadamard product is also known as the elementwise product between two matrices. A nice property of the Hadamard product is that

$$\text{Rank}(P \odot Q) \le \text{Rank}(P) \times \text{Rank}(Q), \tag{1}$$

where $\odot$ denotes the Hadamard product. According to the inequality (1), we can see that the maximal achievable rank of the Hadamard product of two matrices is upper-bounded by the product of their ranks. When $P$ and $Q$ have appropriate sizes to make matrix multiplication feasible, we have

$$\text{Rank}(PQ) \le \min(\text{Rank}(P), \text{Rank}(Q)). \tag{2}$$

Compared the inequalities (1) and (2), we can see that the upper-bound of the rank of the Hadamard product is much larger than that of the matrix multiplication even when $P$ or $Q$ or both have a low rank. Note that the update parameter in LoRA relies on the matrix multiplication of two low-rank matrices and inequality (2) implies that the update parameter in LoRA is low-rank. From this perspective of the upper-bound of the rank, the Hadamard product could help improve that.

The upper-bound of the rank may not precisely reflect the actual rank, but it gives the possible maximal rank of the Hadamard product. To the best of our knowledge, the lower-bound analysis of the rank for the Hadamard product is only for $P$ and $Q$ with special structures (e.g., positive semidefinite matrices (Horn & Yang, 2020)). Empirically we find that the Hadamard product could enhance the rank as demonstrated in Figures 1 and 7.

## 4.3 FORMULATION

Built on the Hadamard product, we give a general formulation for the update parameter matrix as

$$\Delta W = R \odot W_{hi}, \tag{3}$$

where $W_{hi}$ is the trainable parameter and $R$ is a fixed matrix. Based on inequality (1), we obtain

$$\text{Rank}(\Delta W) \le \text{Rank}(R) \times \text{Rank}(W_{hi}). \tag{4}$$

When $R$ has a high rank, inequality (4) suggests that $\Delta W$ can also achieve a high rank, potentially exceeding $\min(d, k)$. To ensure parameter efficiency, we restrict $W_{hi}$ to be low-rank, that is,

$$W_{hi} = AB, \tag{5}$$

where $A \in \mathbb{R}^{d \times r}$, $B \in \mathbb{R}^{r \times k}$, and $r$ is much smaller than $\min(d, k)$. This decomposition defined in Eq. (5) indicates that $W_{hi}$ has a maximum rank of $r$, confirming its low-rank nature.

Based on Eqs. (3) and (5), it is easy to see that when $R$ is chosen to be a matrix of all ones, Eq. (3) gives the formulation of the update parameter matrix in LoRA, making LoRA a special case of HiRA. However, this setting of $R$ is not so informative as it does not utilize any information in the pre-trained LLMs. Instead, in HiRA, we use the frozen parameter $W_0$ in LLMs to be $R$ since $W_0$ could contain useful information of LLMs. By combining all the above considerations together, we can obtain the update parameter for the proposed HiRA method as

$$\Delta W = W_0 \odot (AB). \tag{6}$$

Based on the inequality (4), the rank of $\Delta W$ defined in Eq. (6) is upper-bounded by $r_0 r$, allowing for potentially high rank, as illustrated in Figure 3. Here, $r$ is set equal to the dimension of LoRA, ensuring that the number of trainable parameters remains same and meets the requirements of PEFT.

During the training, $W_0$ remains frozen, while $A$ and $B$ serve as trainable parameters to facilitate the model updating. For a linear layer $h = W_0 x$, HiRA modifies the forward pass of this layer as

$$h = W'x = W_0 x + (W_0 \odot (AB))x.$$

The calculation of $W'$ yields a computational complexity $O(drk + dk) = O(drk)$, which is equivalent to that in LoRA. To ensure that the initial value of update parameters will not modify the original LLMs, we require the initial value of update parameters to be zero matrices. To achieve that, the initial values for $W_{hi}$ could be zero matrix. Under this requirement, $A$ is initialized to be zero matrices, while $B$ is initialized with the kaiming initialization (He et al., 2015).

## 4.4 EFFICIENT MODEL ADAPTATION FOR INFERENCE

During production deployment, HiRA facilitates efficient inference by pre-computing and merging the update parameters into $W_0$ to form $W' = W_0 + W_0 \odot (AB)$. This enables LLMs to switch between tasks swiftly, as the original parameters can be recovered by through element-wise division by $AB + 1$. Then, the LLM can be adapted to new tasks using HiRA. Notably, integrating the update parameters directly into $W_0$ eliminates computational overhead during inference and avoids additional latency commonly associated with other PEFT techniques like Prompt Tuning and P-Tuning. Moreover, MoRA introduces complex mapping functions to compress the input into a relatively high dimension and then decompress back, which cannot be easily merged into the original parameters in LLMs only if the function mappings in the compression and decompression can be represented by a transformation matrix and will incur additionally computational overhead.

## 4.5 RELATIONS WITH INTRINSIC DIMENSIONALITY

Previous studies (Li et al., 2018; Aghajanyan et al., 2021) show that LLMs have a low intrinsic dimensionality, meaning that only a small subset of trainable parameters is necessary for effective fine-tuning. While LoRA (Hu et al., 2021) suggests a low rank for the update parameter matrix $\Delta W$, our findings complement this by showing that the rank of $\Delta W$ can be enhanced under the low intrinsic dimensionality of $W_{hi}$ based on the Hadamard product. It is crucial to differentiate between the intrinsic dimensionality and the rank of $\Delta W$; although a low intrinsic dimensionality implies that only a few parameters need to be fine-tuned, it does not inherently mandate low rank. Our method highlights that increasing the rank of $\Delta W$ is beneficial for enhancing model flexibility and performance, even within a compact parameter space of a low intrinsic dimensionality.

## 4.6 THE EXPRESSIVE POWER OF HIRA

In this section, we analyze the expressive power of HiRA, building on Zeng & Lee (2024), which originally examined the expressive power of LoRA. In HiRA, given the pre-trained weight matrix $W_0$, the updated weight is defined as $W_0 + W_0 \odot (AB)$. We denote by $\overline{E}$ the optimal update and measure the expressive power by the minimal difference between the updated weight and $\overline{E}$. For LoRA with rank $r$, the minimum difference is equal to the $(r + 1)$-th largest singular value, denoted by $\sigma_{r+1}(\overline{E})$. In Theorem 1, we analyze the expressive power of HiRA.

**Theorem 1.** (The Expressive Power of HiRA) *Consider the optimal parameter update $\overline{E}$ and the HiRA update with the form $W_0 \odot W_{hi}$, where the rank of $W_{hi}$ is less than $r$. Then we have*

$$\min_{W_{hi}:\mathrm{Rank}(W_{hi})<r} \left\| W_0 \odot W_{hi} - \overline{E} \right\|_2 \le \sigma_{r+1}(\overline{E} \oslash W_0) \left\| W_0 \right\|_2,$$

*where $\oslash$ denotes the element-wise division.*

According to Theorem 1, the expressive power of HiRA depends on $\overline{E} \oslash W_0$ rather than $\overline{E}$ alone in LoRA. A detailed proof and analysis are provided in Appendix B. The key distinction between HiRA and LoRA lies in the role of $W_0$, whose high-rank properties are inherently tied to the contained information. In this context, $W_0$ in HiRA serves a dual role: it confines and facilitates the adaptation. While the information in $W_0$ limits the flexibility of the update matrix, preventing it

from achieving an unconstrained high-rank matrix, the pre-trained knowledge embedded in $W_0$ also aids the adaptation process, allowing for more efficient fine-tuning.

### 4.7 GRADIENT ANALYSIS

The distinction between HiRA and LoRA also lies in how their gradients interact with the pre-trained weight matrix, $W_0$. Let $y' = W_0 x + \Delta W x$ and $y$ represent the predicted and true labels. For simplicity, we consider a linear neural network with the mean squared error loss $\mathcal{L}$.

In LoRA, the gradients are computed as $\frac{\partial \mathcal{L}}{\partial A} = B^\top (y - y')(-x^\top)$ and $\frac{\partial \mathcal{L}}{\partial B} = (y - y')(-x^\top)A$, which are independent of $W_0$. In contrast, the gradients in HiRA are given by

$$\frac{\partial \mathcal{L}}{\partial A} = B^\top \left( W_0 \odot \left( (y - y')(-x^\top) \right) \right), \quad \frac{\partial \mathcal{L}}{\partial B} = \left( W_0 \odot \left( (y - y')(-x^\top) \right) \right) A.$$

This reveals that HiRA could leverage the information encoded in $W_0$ to guide the adaptation. Thus, HiRA can potentially enhance performance when the pre-trained model has already captured patterns relevant to downstream tasks.

## 5 EXPERIMENT

In this section, we conduct experiments on three tasks to evaluate the proposed HiRA method.

### 5.1 DATASETS

**Commonsense Reasoning.** We utilize eight sub-tasks with predefined training and testing datasets (Hu et al., 2023)[1], combining 170,420 query-answer pairs for fine-tuning LLMs and selecting 120 random entries as a validation set. The sub-tasks include BoolQ (Clark et al., 2019) (yes/no QA), PIQA (Bisk et al., 2020) (physical commonsense), SIQA (Sap et al., 2019) (social reasoning), HellaSwag (Zellers et al., 2019) (commonsense NLI), WinoGrande (Sakaguchi et al., 2021) (fill-in-the-blank), ARC-c and ARC-e (Clark et al., 2018) (multiple-choice science questions), and OBQA (Mihaylov et al., 2018) (multi-step reasoning). Table 11 presents the dataset statistics.

**Open-domain Dialogue Generation.** We use the ConvAI2 dataset (Dinan et al., 2019), including 17,878 training and 1,000 testing multi-turn conversations. Each dialogue features persona profiles of 4–5 descriptive sentences and conversational history. Following (Liu et al., 2020; Song et al., 2021; Huang et al., 2023b;a; 2024), we adopt a self-persona setting, revealing only the speaker's persona.

**Mathematical Reasoning.** For this task, we employ MetaMath (Yu et al., 2023) as the training corpus and GSM8K (Cobbe et al., 2021) as the test dataset.

### 5.2 EXPERIMENTAL SETTINGS

**Evaluation Metric.** To assess the performance on the commonsense reasoning datasets, we employ accuracy as the primary metric for each sub-task. For each test instance, the language models decode answers from the provided queries. We then search for the presence of specific answer keywords (e.g., "true" or "false" for BoolQ). The first occurrence of the keyword is recorded as the model's response. If no relevant keywords are identified, the model is considered to have failed to correctly answer the commonsense reasoning question. This method allows us to consistently evaluate the performance of model responses across all eight sub-tasks, and is adopted by (Hu et al., 2023; Liu et al., 2024). For the CONVAI2 dataset, we use the BLEU (Papineni et al., 2002) and BERT Score (Zhang et al., 2019) as the evaluation metrics. For the mathematical reasoning task, we utilize the accuracy as the evaluation metric.

**Baseline Methods.** We compare the HiRA with prompt-based methods including Prompt Tuning (Lester et al., 2021) and P-tuning (Liu et al., 2022), and low-rank adaptation methods including LoRA (Hu et al., 2021), DoRA (Liu et al., 2024), and MoRA (Jiang et al., 2024). Our experiments use the Llama-2-7B (Touvron et al., 2023) and Llama-3-8B (Dubey et al., 2024) open-source LLMs.

---

[1] https://github.com/AGI-Edgerunners/LLM-Adapters/tree/main/dataset

Table 1: Accuracy comparison among various PEFT methods on commonsense reasoning datasets. Results for ChatGPT, LoRA, and DoRA are sourced from (Liu et al., 2024). The best performance within each LLM is indicated in **bold**, while the second best performance is highlighted in underline.

| Model | Method | Params (%) | BoolQ | PIQA | SIQA | ARC-c | ARC-e | OBQA | HellaS | WinoG | Average |
|---|---|---|---|---|---|---|---|---|---|---|---|
| ChatGPT | - | - | 73.10 | 85.40 | 68.50 | 79.90 | 89.80 | 74.80 | 78.50 | 66.10 | 77.01 |
| Llama-2-7B | Prompt Tuning | 0.0012 | 55.93 | 12.35 | 30.50 | 6.06 | 8.63 | 9.40 | 6.91 | 40.57 | 21.29 |
| | P-Tuning | 0.7428 | 58.75 | 36.02 | 0.20 | 0.17 | 1.98 | 0.80 | 0.01 | 0.00 | 12.24 |
| | LoRA ($r = 32$) | 0.8256 | 69.80 | 79.90 | 79.50 | 64.70 | 79.80 | 81.00 | 83.60 | 82.60 | 77.61 |
| | DoRA ($r = 32$) | 0.8256 | 71.80 | **83.70** | 76.00 | 68.20 | 83.70 | 82.40 | **89.10** | 82.60 | 79.69 |
| | MoRA ($r = 32$) | 0.8241 | **72.17** | 80.79 | **79.53** | 71.42 | 85.31 | 81.20 | 29.09 | 80.19 | 72.46 |
| | HiRA ($r = 16$) | 0.4128 | 69.82 | 80.20 | 78.20 | 71.33 | 85.90 | 81.00 | 86.99 | 83.43 | 79.61 |
| | HiRA ($r = 32$) | 0.8256 | 71.22 | 83.35 | **79.53** | **73.81** | **86.74** | **84.60** | 88.12 | **83.98** | **81.42** |
| Llama-3-8B | Prompt Tuning | 0.0010 | 56.85 | 45.05 | 36.13 | 31.57 | 32.74 | 29.20 | 14.01 | 50.12 | 36.96 |
| | P-Tuning | 0.6240 | 59.97 | 11.64 | 8.19 | 7.42 | 8.63 | 9.60 | 1.77 | 37.65 | 18.11 |
| | LoRA ($r = 32$) | 0.7002 | 70.80 | 85.20 | 79.90 | 71.20 | 84.20 | 79.00 | 91.70 | 84.30 | 80.79 |
| | DoRA ($r = 32$) | 0.7002 | 74.60 | 89.30 | 79.90 | 80.40 | 90.50 | 85.80 | 95.50 | 85.60 | 85.20 |
| | MoRA ($r = 32$) | 0.6997 | 74.28 | 87.43 | 80.71 | 79.61 | 91.16 | 85.60 | 43.53 | 86.74 | 78.63 |
| | HiRA ($r = 16$) | 0.3513 | 73.85 | 89.12 | 81.06 | 82.59 | 93.06 | 87.40 | 94.85 | 86.74 | 86.08 |
| | HiRA ($r = 32$) | 0.7002 | **75.40** | **89.70** | **81.15** | **82.90** | **93.27** | **88.32** | **95.36** | **87.70** | **86.72** |

**Implementation Details.** Following the identical training setup to (Liu et al., 2024) except learning rate adjustments, we implement HiRA on the Llama-2-7B and Llama-3-8B models with $r = 16$ and $r = 32$, respectively. The AdamW optimizer (Loshchilov & Hutter, 2019) is employed with a learning rate 0.001, which warms up for 100 steps. For the commonsense reasoning dataset, we fine-tune LLMs for 3 epochs, with evaluations at every 80 step to select the best checkpoint based on the validation set. We place LoRA, DoRA, MoRA and HiRA on the query, key, value weights, and two linear layers (i.e., down and up projection) in attention modules. To ensure fair comparisons among LoRA, DoRA, MoRA, and HiRA, we maintain the same or comparable numbers of trainable parameters. For Prompt Tuning and P-Tuning, which inherently involve fewer trainable parameters due to their reliance on prefix soft prompts, we adjust accordingly to keep the number of trainable parameters comparable. The detail can be found in Appendix A.1. HiRA is evaluated over 5 runs with different random seeds. Experiments on the CONVAI2 dataset use 1 training epoch, while the mathematical reasoning task uses 2 epochs, which is the only difference from the settings above.

## 5.3 RESULTS ON COMMONSENSE REASONING TASKS

As shown in Table 1, HiRA consistently outperforms all baseline methods in terms of average accuracy across both the Llama-2-7B and Llama-3-8B models. For $r = 32$, HiRA achieves an average accuracy of 81.42% for the Llama-2-7B model, surpassing the best baseline, DoRA, which records 79.69%. In the case of the Llama-3-8B model, HiRA shows a significant improvement in terms of the average accuracy (86.72% vs. 85.20%) over the best-performing baseline (i.e., DoRA). These results underscore HiRA's effectiveness in leveraging the Hadamard product to enhance the model capacity and performance within the PEFT strategy.

In contrast, for $r = 16$, HiRA achieves an average accuracy of 79.61% for Llama-2-7B model, utilizing half the number of trainable parameters compared to LoRA and DoRA while achieving comparable or even performance. Similarly, for the Llama-3-8B model, HiRA with $r = 16$ achieves an impressive average accuracy of 86.08%. This demonstrates HiRA's capability to deliver strong performance while maintaining a lower intrinsic dimensionality.

## 5.4 RESULTS ON CONVERSATIONAL TASK

According to the results on the CONVAI2 dataset as shown in Table 2, HiRA consistently outperforms all baseline methods across all comparison metrics. Specifically, HiRA with $r = 32$ achieves the highest average score of 47.80%, closely followed by HiRA with $r = 16$, which also performs exceptionally well with an average score of 47.79% despite using only half of the number of trainable parameters in LoRA and its variants. Both configurations of HiRA surpass DoRA and LoRA, which exhibit similar results with average scores of 46.62% and 46.59%, respectively. MoRA, while not as strong as LoRA in this task, still outperforms Prompt Tuning. Those results further substantiate the superiority of HiRA in both parameter efficiency and performance across various tasks.

Table 2: Results on the CONVAI2 dataset, where BERT F1, BERT-R, and BERT-P denote the F1, Precision, and Recall based on the BERT score, respectively.

| Model | Method | Params (%) | BLEU | BERT F1 | BERT-R | BERT-P | Meteor | R-L | Average |
|-------|--------|-----------|------|---------|--------|--------|--------|-----|---------|
| | Prompt Tuning | 0.0012 | 0.04 | 72.44 | 77.38 | 68.23 | 0.80 | 0.80 | 36.62 |
| | P-Tuning | 0.7428 | 0.60 | 83.29 | 83.33 | 83.28 | **15.11** | 12.36 | 46.33 |
| | MoRA ($r = 32$) | 0.8241 | 1.09 | 84.09 | 84.65 | 83.59 | 10.97 | 9.57 | 45.66 |
| Llama-2-7B | LoRA ($r = 32$) | 0.8256 | 1.82 | 84.41 | 84.71 | 84.16 | 11.38 | 10.55 | 46.17 |
| | DoRA ($r = 32$) | 0.8256 | 1.73 | 84.18 | 84.61 | 83.81 | 11.25 | 10.41 | 46.00 |
| | HiRA ($r = 16$) | 0.4128 | 2.56 | 83.97 | 84.12 | 83.86 | 13.35 | 12.58 | 46.74 |
| | HiRA ($r = 32$) | 0.8256 | **2.70** | **84.86** | **84.98** | **84.77** | 13.56 | **12.80** | **47.28** |
| | Prompt Tuning | 0.0012 | 1.45 | 82.99 | 82.99 | 83.05 | 14.72 | 13.13 | 46.39 |
| | P-Tuning | 0.7428 | 1.50 | 81.52 | 81.07 | 82.01 | **15.49** | 13.55 | 45.86 |
| | MoRA ($r = 32$) | 0.8241 | 1.60 | 84.22 | 84.06 | 84.43 | 12.37 | 11.19 | 46.31 |
| Llama-3-8B | LoRA ($r = 32$) | 0.8256 | 2.26 | 84.32 | 84.00 | 84.67 | 12.51 | 11.77 | 46.59 |
| | DoRA ($r = 32$) | 0.8256 | 2.29 | 84.32 | 84.06 | 84.62 | 12.63 | 11.78 | 46.62 |
| | HiRA ($r = 16$) | 0.4128 | 3.32 | **84.84** | **84.41** | **85.30** | 14.93 | 13.94 | 47.79 |
| | HiRA ($r = 32$) | 0.8256 | **3.41** | 84.81 | 84.40 | 85.25 | 14.87 | **14.05** | **47.80** |

## 5.5 RESULTS ON MATHEMATICAL REASONING TASKS

We evaluate the performance of HiRA on mathematical reasoning tasks using the MetaMath dataset for training and the GSM8K benchmark for evaluation. As shown in Table 3, HiRA significantly outperforms baseline methods, achieving an accuracy of 70.81%. This represents a notable improvement over LoRA (65.89%), DoRA (66.12%), and MoRA (67.98%). Even with fewer trainable parameters ($r = 16$), HiRA remains competitive, achieving 67.63%. On the Llama-2-7B model, HiRA delivers strong results, achieving 46.85%, a substantial increase compared to LoRA (15.16%) and its variants.

Table 3: Results on mathematical reasoning tasks.

| Model | Method | Trainable | GSM8K |
|-------|--------|-----------|-------|
| | Prompt Tuning | 0.0012 | 15.62 |
| | P-Tuning | 0.7428 | 2.65 |
| | LoRA ($r = 32$) | 0.7002 | 65.89 |
| Llama-3-8B | DoRA ($r = 32$) | 0.7002 | 66.12 |
| | MoRA ($r = 32$) | 0.6997 | 67.98 |
| | HiRA ($r = 16$) | 0.4128 | 67.63 |
| | HiRA ($r = 32$) | 0.7002 | **70.81** |

These results demonstrate HiRA's superior ability to adapt to complex mathematical reasoning, attributed to its high-rank updates, which enhance the model's expressive power while maintaining parameter efficiency.

## 6 ABLATION STUDIES

### 6.1 SINGULAR VALUE ANALYSIS OF FULL FINE-TUNING, LORA AND HIRA

To further analyze the advantages of HiRA, we calculate the singular values of update parameter matrix in full fine-tuning (FFT), LoRA, and HiRA. In Figure 4, we present the number of singular

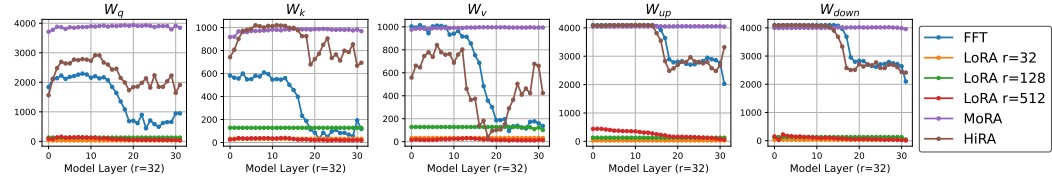

Figure 4: Count of singular values exceeding 0.005 across layers for FFT, LoRA, MoRA, and HiRA.

values exceeding 0.005 for each layer. FFT produces diverse singular value distributions across different layers, reflecting the difference in the ease of updating layers for fine-tuning. Notably, HiRA exhibits similar trends to those of FFT, indicating its higher rank is crucial for effective optimization. In contrast, the inherently low-rank structure of LoRA may limit its expressive power.

Figure 5 presents the sum of the squared singular values per layer, which corresponds to the squared Frobenius norm. The results indicate that MoRA and LoRA with rank=32 yields substantially larger singular values, as reported in (Lialin et al., 2024). Those large singular values suggest that its updates make strong adjustments in certain directions on the pre-trained model to swiftly adapt to new tasks. However, excessively large singular values (i.e., the spectral norm) may increase the risk of overfitting, consequently impairing the generalization ability (Bartlett et al., 2017). Moreover,

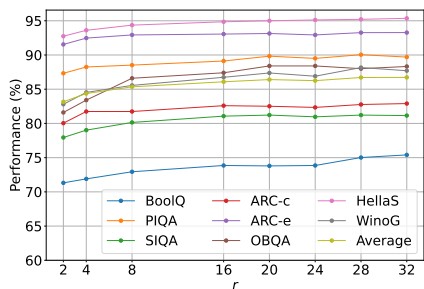

Figure 5: Sum of squared singular values across layers for FFT, LoRA, MoRA, and HiRA.

overly large singular values can induce gradient explosion or vanishing issues, thereby compromising the stability and convergence rate of subsequent training (Pennington et al., 2017). In contrast, HiRA tends to maintain smaller singular values as FFT did.

## 6.2 IMPACT OF $r$ ON MODEL PERFORMANCE

Figure 6 illustrates the impact of the parameter dimension $r$ to the model performance across multiple commonsense reasoning tasks. As $r$ increases from 2 to 32, there is a consistent and notable improvement in performance, with the average accuracy rising from $83.16\%$ to $86.72\%$. This trend underscores the importance of a higher-rank structure in enhancing the model ability to generalize across diverse tasks. Notably, tasks such as PIQA and HellaSwag exhibit significant gains, with PIQA improving by $2.37\%$ as $r$ increases, highlighting the critical role of $r$ in tasks that require advanced reasoning capabilities. Interestingly, even at $r = 8$, HiRA still outperforms LoRA with $r = 32$, demonstrating the superiority of HiRA to efficiently leverage a smaller number of trainable parameters to achieve good performance, making HiRA a compelling choice even under constrained resources.

Figure 6: Performance of HiRA across tasks when $r$ increases.

## 6.3 IMPACT OF DIFFERENT CHOICES OF $R$

Table 4: Performance comparison between different choices of $R$ defined in Eq. (3). $HiRA_{rand}$ denotes a variant using randomly initialized $R$ instead of $W_0$. HiRA denotes the vanilla version where $R$ equals $W_0$. Both methods are tuned on the Llama-3-8B model.

| Model | BoolQ | PIQA | SIQA | ARC-c | ARC-e | OBQA | HellaS | WinoG | Average |
|---|---|---|---|---|---|---|---|---|---|
| HiRA | **75.40** | **89.70** | **81.15** | **82.90** | **93.27** | **88.32** | **95.36** | **87.70** | **86.72** |
| $HiRA_{rand}$ | 62.17 | 50.38 | 33.62 | 26.62 | 26.39 | 26.40 | 25.06 | 50.36 | 37.63 |

In this section, we explore the impact of different choices for $R$ in Eq. (3) on performance in commonsense reasoning tasks. Specifically, we compare with a variant, $HiRA_{rand}$, where $R$ is randomly generated from a uniform distribution $[0, 1]$ before the training and remains fixed. As shown in Table 4, both methods use identical training protocols by utilizing the same optimizer, learning rate, and training epochs, yet HiRA significantly outperforms $HiRA_{rand}$. This highlights the effectiveness of using $W_0$ as $R$. Moreover, as discusses in Section 4.4, using $W_0$ for $R$ aids in recovering $W_0$ from the merged parameters $W'$ given $A$ and $B$, whereas $HiRA_{rand}$ requires storing separately to achieve that, which could lead to additional storage costs.

## 6.4 RANK ANALYSIS

In this section, we compare the average rank of the update parameter matrix $\Delta W$ over layers for HiRA, LoRA, and MoRA, which have comparable numbers of trainable parameters. As shown in Figure 7, HiRA possess $\Delta W$ with much higher ranks than LoRA and MoRA, indicating that HiRA can achieve high-rank adaptation under the PEFT strategy via the Hadamard product. Notably, as the layer goes deeper, the rank of $\Delta W$ first increases and then fluctuates, which indicates that deeper layers may need a higher-rank $\Delta W$ to adapt to new tasks. Overall, HiRA attains higher-rank $\Delta W$'s across all layers, which correlates with improved performance as detailed in Table 1.

Table 5: Performance of the Llama-3-8B model with HiRA integrated into various components.

| Component | BoolQ | PIQA | SIQA | ARC-c | ARC-e | OBQA | HellaS | WinoG | Average |
|---|---|---|---|---|---|---|---|---|---|
| FC, QKV | **75.40** | **89.70** | 81.15 | **82.90** | **93.27** | **88.32** | **95.36** | **87.70** | **86.72** |
| FC | 73.30 | 89.45 | **81.17** | 82.51 | 92.89 | 87.60 | 94.82 | 86.58 | 86.04 |
| QV | 73.09 | 88.85 | 81.06 | 80.38 | 92.68 | 86.20 | 94.37 | 85.87 | 85.31 |
| QKV | 72.26 | 89.23 | 80.30 | 80.89 | 92.97 | 84.80 | 94.37 | 86.19 | 85.13 |
| QK | 71.38 | 87.49 | 78.92 | 80.80 | 91.29 | 83.00 | 93.68 | 84.37 | 83.86 |
| V | 67.46 | 87.60 | 79.48 | 79.01 | 91.20 | 82.20 | 92.91 | 83.58 | 82.93 |
| Q | 68.59 | 86.51 | 77.64 | 79.61 | 90.99 | 81.20 | 92.68 | 81.93 | 82.39 |
| K | 68.20 | 86.07 | 77.74 | 80.03 | 90.82 | 80.00 | 92.29 | 81.37 | 82.07 |

## 6.5 ANALYSIS ON PLACEMENT OF HiRA IN TRANSFORMERS

As shown in Table 5, we analyze the effectiveness of HiRA when applied to any subset of weight matrices within the Transformer, including fully connected (FC) layers and query (Q), key (K), and value (V) in attention module, across multiple commonsense reasoning tasks. The results indicate that applying HiRA to the FC layers and the combination of QKV matrices yields the best performance across most tasks. In contrast, applying HiRA to individual components leads to inferior performance. Hence, we choose to apply HiRA on both FC layers and the combined QKV matrices.

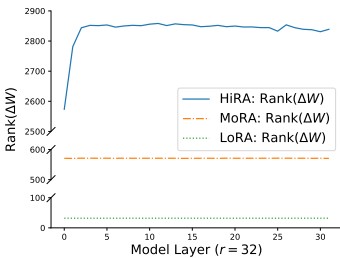

Figure 7: Average rank of $\Delta W$ for HiRA, MoRA, and LoRA tuned on the Llama-3-8B.

Table 6: The results of the HiLoRA, where the Llama-3-8B model is used. The best performance is shown in **bold**, while the second best performance is highlighted in underline.

| Model | BoolQ | PIQA | SIQA | ARC-c | ARC-e | OBQA | HellaS | WinoG | Average |
|---|---|---|---|---|---|---|---|---|---|
| LoRA ($r = 32$) | 70.80 | 85.20 | 79.90 | 71.20 | 84.20 | 79.00 | 91.70 | 84.30 | 80.79 |
| HiRA ($r = 32$) | 75.40 | 89.70 | 81.15 | 82.90 | 93.27 | 88.32 | 95.36 | 87.70 | 86.72 |
| HiLoRA ($r_1 = 2, r_2 = 30$) | 73.76 | 84.98 | 80.60 | 74.57 | 88.38 | 84.60 | 88.68 | 86.27 | 82.73 |
| HiLoRA ($r_1 = 4, r_2 = 28$) | 74.22 | 90.26 | 82.80 | 82.25 | 93.48 | **89.80** | 96.13 | 88.48 | 87.18 |
| HiLoRA ($r_1 = 8, r_2 = 24$) | 73.52 | 89.77 | 82.65 | 83.36 | **93.81** | 89.60 | 96.04 | 89.03 | 87.22 |
| HiLoRA ($r_1 = 16, r_2 = 16$) | 74.77 | 89.88 | 81.73 | 83.28 | 92.93 | 88.20 | 95.52 | 88.08 | 86.80 |
| HiLoRA ($r_1 = 20, r_2 = 12$) | **75.84** | **90.42** | 82.14 | **84.22** | 93.73 | 87.80 | **96.45** | 89.27 | **87.48** |
| HiLoRA ($r_1 = 24, r_2 = 8$) | 74.89 | 89.61 | 81.68 | 82.85 | 92.93 | 86.20 | 95.68 | **89.34** | 86.65 |
| HiLoRA ($r_1 = 28, r_2 = 4$) | 73.64 | 88.96 | 81.06 | 83.11 | 93.14 | 87.80 | 95.51 | 88.71 | 86.49 |
| HiLoRA ($r_1 = 30, r_2 = 2$) | 73.46 | 89.50 | **83.06** | 82.34 | 93.10 | 89.00 | 95.86 | 88.87 | 86.90 |

## 6.6 COMBINATION WITH LoRA

In this section, we explore the combination of HiRA with LoRA to investigate the impact of integrating both techniques into the performance. Specifically, such combination, which is called **HiLoRA** in this section for clarity, formulates the update parameter matrix as $\Delta W = W_0 \odot AB + L_1 L_2$, where $A \in \mathbb{R}^{d \times r_1}$, $B \in \mathbb{R}^{r_1 \times k}$, $L_1 \in \mathbb{R}^{d \times r_2}$, and $L_2 \in \mathbb{R}^{r_2 \times k}$. It is easy to see that when $r_2$ becomes 0, HiLoRA degenerates to HiRA, and if $r_1$ equals 0, HiLoRA becomes LoRA.

To compare HiLoRA with HiRA and LoRA fairly, we ensure the same numbers of trainable parameters by setting $r$ in HiRA and LoRA to $r_1 + r_2$. Table 6 presents the results of HiLoRA with various configurations on $r_1$ and $r_2$. The results indicate that increasing $r_1$ generally improves performance. HiLoRA with $r_1 = 20$ and $r_2 = 12$ achieves the highest average score, excelling in tasks like HellaSwag and ARC-c. This suggests that a higher intrinsic dimension for HiRA is preferable over LoRA in HiLoRA, which demonstrates the usefulness of HiRA that could have larger capacity. Moreover, most configurations of $r_1$ and $r_2$ perform comparably or better than HiRA and LoRA with $r = 32$, indicating that the combination of LoRA and HiRA is promising for fine-tuning LLMs.

## 7 CONCLUSION

In this paper, we introduced HiRA, a novel high-rank adaptation that maintains comparable numbers of trainable parameters while enhancing the rank of update parameters. HiRA offers a cost-effective alternative to LoRA, providing similar benefits but without additional inference overhead. Extensive experiments demonstrate the effectiveness of the HiRA method. In our future work, we are interested in applying HiRA to more applications.

## ACKNOWLEDGEMENTS

This work is supported by National Key R&D Program of China 2022ZD0160300 and NSFC key grant under grant no. 62136005.

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

# A TRAINING DETAILS

## A.1 TRAINING HYPERPARAMETERS

Table 7 outlines hyperparameters used for tuning the Llama-2-7B and Llama-3-8B models with HiRA across two tasks: commonsense reasoning and CONVAI2. Both tasks utilize the same hyper-parameter set, with the primary distinction being the number of epochs; the commonsense reasoning task runs for three epochs, whereas CONVAI2 runs for a single epoch. Each experiment is conducted separately, employing varying $r$ for a single run on each model. The best results are chosen based on the validation loss. For baseline methods, the hyperparameters also reuse the above configurations. Different PEFT methods could have different hyperparameters to be set for comparable or the same number of trainable parameters.

Table 7: Hyperparameters for HiRA.

| Hyperparameter | Value |
|---|---|
| Optimizer | AdamW |
| Weight Decay | 0 |
| Base Model | [Llama-2-7B, Llama-3-8B] |
| Learning Rate | [0.0001, 0.0002] |
| r | $[2, 4, 8, 16, 24, 28, 30, 32]$ |
| Warm Up | 100 steps |
| Batch Size | 32 |
| Target Modules | q_proj, k_proj, v_proj, up_proj, down_proj |
| Evaluation Steps | Every 80 steps |

Table 8: The statistics of the CONVAI2 dataset.

| Data Split | Utterances | Dialogues | Personas |
|---|---|---|---|
| Train | 131,438 | 17,878 | 1,155 |
| Test | 7,801 | 1,000 | 100 |

## A.2 STATISTICS OF THE CONVAI2 DATASET

We conduct experiments using the ConvAI2 dataset (Dinan et al., 2019), a benchmark for open-domain dialogue generation tasks. As shown in Table 8, this dataset consists of 17,878 training and 1,000 testing multi-turn conversations collected from crowdworkers. Each dialogue features persona profiles that provide four to five sentences describing each speaker's background, along with the conversational history between the two interlocutors. In our experiments, by following (Liu et al., 2020; Song et al., 2021; Huang et al., 2023b;a), we utilize a self-persona setting, where only the speaking interlocutor's persona is revealed, while the persona of the other remains undisclosed.

## A.3 EXPERIMENTAL SETTINGS FOR LoRA WITH DIFFERENT RANKS

In this section, we detail the experimental settings used for fine-tuning the Llama-3-8B model with various LoRA configurations. The specific hyperparameters, including the rank ($r$), learning rate (lr), and batch size, are outlined in Table 9.

Table 9 lists the specific configurations used for each LoRA rank. Higher ranks, such as $r = 1024$, required a lower learning rate of $1.00e - 06$ and a smaller batch size of 8 to prevent issues like gradient explosion. In contrast, lower ranks like $r = 32$ could be trained with a higher learning rate of $1.00e - 04$ and a larger batch size of 72.

Table 10 provides the general hyperparameters that were consistent across all experiments, ensuring a fair comparison between different configurations. These settings were designed to optimize the fine-tuning process for the Llama-3-8B model while maintaining computational efficiency.

Table 9: LoRA Configurations with Different Ranks, Learning Rates, and Batch Sizes.

| LoRA Config | Learning Rate (lr) | Batch Size |
|---|---|---|
| r=1024 | 1.00e-06 | 8 |
| r=512 | 1.00e-06 | 16 |
| r=256 | 1.00e-05 | 32 |
| r=128 | 5.00e-05 | 72 |
| r=32 | 1.00e-04 | 72 |

Table 10: General Hyperparameters Used Across All Experiments.

| Hyperparameter | Value |
|---|---|
| Optimizer | AdamW |
| Weight Decay | 0 |
| Base Model | Llama-3-8B |
| Warm Up | 100 steps |
| Target Modules | q_proj, k_proj, v_proj, up_proj, down_proj |
| Evaluation Steps | Every 80 steps |

## A.4 TRAINING COST

The computational cost for training on the commonsense reasoning task requires 14 GPU hours over 3 epochs on Nvidia-A100 80G GPU on Llama-3-8B, while the CONVAI2 task requires 9 GPU hours for a single epoch under the HiRA ($r = 32$) on Nvidia-A100 80G on Llama-3-8B.

## A.5 STATISTICS OF COMMONSENSE REASONING DATASET

As illustrated in Table 11, the training set comprises a mix of eight sub-tasks totalling $170,300$ entries, while the validation set contains a random selection of $120$ entries. The test dataset also covers these eight sub-tasks and is evaluated using a single trained model.

Table 11: The detailed statistics of commonsense reasoning datasets.

| Dataset | Data Number | Type |
|---|---|---|
| Train | 170,300 | Mixed |
| Validation | 120 | Mixed |
| Test | | |
| BoolQ | 3,270 | Yes/No |
| PIQA | 1,830 | Option |
| SIQA | 1,954 | Option |
| HellaSwag | 10,042 | Option |
| WinoGrande | 1,267 | Option |
| ARC-e | 2,376 | Option |
| ARC-c | 1,172 | Option |
| OBQA | 500 | Option |

## A.6 GPU MEMORY CONSUMPTION AND RUNNING TIME ANALYSIS

This section presents a comparative analysis of GPU memory consumption and running time between the LoRA method and the proposed HiRA configurations under similar settings of trainable parameters. The experiments were conducted using the LLaMA-3-8B model.

Table 12: Comparison of GPU Memory Consumption and Running Time

| Configuration | GRAM (GB) | Training Hours |
|---|---|---|
| LoRA ($r = 32$) | 65.48 | 15 hours 0 minutes 56 seconds |
| HiRA ($r = 32$) | 61.49 | 14 hours 09 minutes 32 seconds |

EXPERIMENTAL SETUP

- **Model:** LLaMA-3-8B
- **Batch Size:** 72
- **Epochs:** 3
- **Data Size:** 170k entries
- **LoRA Rank:** 32
- **HiRA Rank:** 32

## B  PROOF OF HiRA'S EXPRESSIVE POWER

In this section, we give the details proof of the expressive power of HiRA in comparison to LoRA. We begin by introducing the Eckart-Young-Mirsky Theorem (Eckart & Young, 1936), which provides the optimal low-rank approximation of a matrix. We will refer to this theorem as Lemma 1.

**Lemma 1.** (Eckart-Young-Mirsky Theorem) *The best rank-$r$ approximation of a matrix $W$ in the spectral norm is given by the $(r + 1)$-th largest singular value, i.e.,*

$$\min_{\hat{W}:Rank(\hat{W})<r} \left\| W - \hat{W} \right\|_2 = \sigma_{r+1}(W).$$

According to Theorem 5.5.1 in (Horn & Johnson, 1994), we have the following Lemma 2.

**Lemma 2.** *For matrices $A$ and $B$, the Hadamard product satisfies $\|A \odot B\|_2 \leq \|A\|_2 \|B\|_2$.*

**Theorem 1.** (The Expressive Power of HiRA) *Consider the optimal parameter update $\overline{E}$ and the HiRA update with the form $W_0 \odot W_{hi}$, where the rank of $W_{hi}$ is less than $r$. Then we have*

$$\min_{W_{hi}:\text{Rank}(W_{hi})<r} \left\| W_0 \odot W_{hi} - \overline{E} \right\|_2 \leq \sigma_{r+1}(\overline{E} \oslash W_0) \|W_0\|_2 ,$$

*where $\oslash$ denotes the element-wise division.*

*Proof.* The proof proceeds as follows

$$
\begin{aligned}
\min_{W_{hi}} \left\| W_0 \odot W_{hi} - \overline{E} \right\|_2 &= \min_{W_{hi}} \left\| W_0 \odot (W_{hi} - \overline{E} \oslash W_0) \right\|_2 \\
&\leq \min_{W_{hi}} \|W_0\|_2 \left\| (W_{hi} - \overline{E} \oslash W_0) \right\|_2 \\
&= \sigma_{r+1}(\overline{E} \oslash W_0) \|W_0\|_2 .
\end{aligned}
$$

In this analysis, we assume that $W_0$ contains no zeros and use $\oslash$ to denote the element-wise division. The optimization objective $\min_{W_{hi}} \left\| W_0 \odot (W_{hi} - \overline{E} \oslash W_0) \right\|_2$ is equivalent to the weighted low-rank approximation problem, which is NP-hard (Srebro & Jaakkola, 2003). As a result, we can analyze it by applying the upper bound provided by Lemma 2. This indicate that HiRA's expressive power is linked to the singular value of $\overline{E} \oslash W_0$.  □

Notably, HiRA can exhibit greater expressive power than LoRA, particularly since the updated weights correspond to the original weights. For instance, consider the following matrices

$$\overline{E} = \begin{bmatrix} 2 & 1 \\ 1 & 2 \end{bmatrix}, \quad W = \begin{bmatrix} 4 & 2 \\ 1 & 2 \end{bmatrix}, \quad \overline{E} \oslash W = \begin{bmatrix} 0.5 & 0.5 \\ 1 & 1 \end{bmatrix}.$$

In this toy example, we find that $\sigma_{r+1}(\overline{E} \oslash W) = 0$, while $\sigma_{r+1}(\overline{E}) = 1$. Thus we have $\sigma_{r+1}(\overline{E} \oslash W) \|W\|_2 < \sigma_{r+1}(\overline{E})$.

## C    COMPARISON OF EFFECTIVE RANK WITH DIFFERENT METHODS

We conducted additional experiments to analyze the effective rank (Roy & Vetterli, 2007) of different fine-tuning methods, as presented in Figure 8.

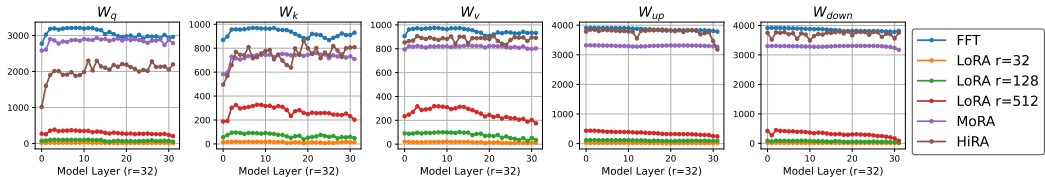

Figure 8: Effective rank across layers for FFT, LoRA, MoRA, and HiRA.

Unlike simply calculating the rank or counting the number of singular values above a given threshold, the effective rank accounts for the full singular value spectrum. For a matrix $A$ of size $M \times N$ with $K = \min\{M, N\}$, the effective rank is defined as follows:

$$\text{erank}(A) = \exp\left(-\sum_{k=1}^{Q} p_k \log p_k\right), \quad p_k = \frac{\sigma_k}{\|\sigma\|_1},$$

where $\sigma_k$ denotes the $k$-th largest singular value and $\|\cdot\|_1$ is the $l_1$-norm of the singular values.

From the results, we can observe that the update matrix obtained by FFT achieves the highest effective rank. For the query component, MoRA shows behavior closer to FFT, while for the value component—and especially for the projection layer (up and down components)—HiRA exhibits better. Conversely, even LoRA with rank=512 offers a relatively lower effective rank. This analysis follows a similar approach to that in (Shuttleworth et al., 2024).

## D    EVALUATION ON TRANSFER LEARNING TASKS

We trained HiRA and LoRA separately on eight commonsense reasoning tasks and evaluated their performance on these tasks. As shown in the Figure 9, HiRA demonstrates strong transferability across tasks, achieving high scores on related tasks. This indicates its ability to generalize effectively beyond the specific task it was trained on.

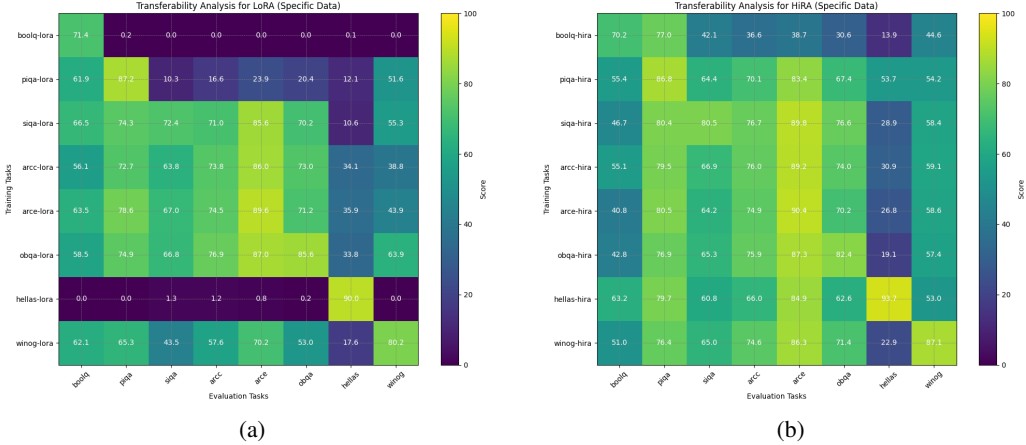

Figure 9: Transferability analysis for (a) LoRA and (b) HiRA. Each row represents the performance of a fine-tuned model trained on a specific task dataset when evaluated across eight different tasks.

In comparison, LoRA exhibits more pronounced overfitting. While it performs well on its training task (e.g., boolq-lora scores 71.44 on boolq), its performance drops significantly on other tasks. In

contrast, HiRA maintains respectable transferability, as seen with arce-hira, which achieves 64.23 on siqa, 70.20 on obqa, and 58.56 on winog, outperforming LoRA in these cases. HiRA also excels at harder tasks requiring more complex reasoning, such as HellaSwag (hellas) and WinoGrande (winog). On the other hand, LoRA performs relatively better on simpler tasks like BoolQ and PIQA, but its overfitting tendency causes significant drops in performance on unrelated tasks.

In summary, HiRA shows superior generalization and is less prone to overfitting compared to LoRA. Evaluating HiRA on transfer learning tasks could further validate its robustness and adaptability.

# E IMPACT OF MORE CHOICES OF $R$

## E.1 REDUCING THE RANK OF $R$

To validate the correlation between the rank of $\Delta W$ and expressiveness, we conducted an additional experiment by reducing the rank of $R$ on LLaMA-3-8B across commonsense reasoning tasks. Specifically, $R$ is derived from $W_0$, the pretrained weight matrix, using SVD decomposition. We retained only the top-$v$ singular values and the corresponding singular vectors to form a reduced-rank version, $W_{0_{[:v]}}$. The results, presented in Table 13, show that increasing the rank of $R$ generally improves the performance across tasks.

Table 13: Performance comparison between $R$ with different ranks based on $W_0$.

| Model | Rank | BoolQ | PIQA | SIQA | ARC-c | ARC-e | OBQA | HellaS | WinoG | Average |
|---|---|---|---|---|---|---|---|---|---|---|
| $R = W_{0,[:32]}$ | 32 | 73.67 | 88.63 | 80.04 | 80.97 | 93.18 | 86.00 | 94.31 | 86.58 | 85.42 |
| $R = W_{0,[:128]}$ | 128 | 74.16 | 89.23 | 82.04 | 80.89 | 93.10 | 87.20 | 95.11 | 87.45 | 86.15 |
| $R = W_{0,[:512]}$ | 512 | 74.34 | **89.88** | **81.99** | 82.25 | **93.86** | 88.20 | **95.38** | 87.61 | 86.69 |
| $R = W_0$ (Ours) | 2852 | **75.40** | 89.70 | 81.15 | **82.90** | 93.27 | **88.32** | 95.36 | **87.70** | **86.72** |

## E.2 USING $W_0$'S IN ADJACENT LAYERS AS $R$

We experimented with a smoothed version of $W_0$ by incorporating information from adjacent layers. Specifically, the smoothed $W_0$ ($R = W_{smooth}$) was computed as the average of the upper, current, and lower layers for the same transformer component. The results, summarized in the Table 14, show that the smoothed $W_0$ achieves an average score of 83.71, slightly lower than the original $W_0$, which achieves 86.72. While smoothing provides competitive performance, the original $W_0$ retains a clear advantage, likely due to its direct alignment with the pretrained weights, preserving task-specific expressiveness and parameter efficiency.

Table 14: Performance comparison between smoothed $W_0$ and $W_0$ used as $R$.

| Model | BoolQ | PIQA | SIQA | ARC-c | ARC-e | OBQA | HellaS | WinoG | Average |
|---|---|---|---|---|---|---|---|---|---|
| $R = W_{smooth}$ | 72.02 | 83.84 | 79.84 | 81.23 | 90.24 | 85.60 | 92.07 | 84.85 | 83.71 |
| $R = W_0$ (Ours) | **75.40** | **89.70** | **81.15** | **82.90** | **93.27** | **88.32** | **95.36** | **87.70** | **86.72** |

# F EXPERIMENTAL EVIDENCE ON HIRA'S EXPRESSIVE POWER

Theorem 1 establishes an upper bound on the approximation error of HiRA, which is proportional to $\sigma_{r+1}(\overline{E} \oslash W_0)$, in contrast to the $\sigma_{r+1}(\overline{E})$ in LoRA. In this section, we use experimental evidence to prove the advantages of $\sigma_{r+1}(\overline{E} \oslash W_0)$.

Since the optimal update matrix $\overline{E}$ comes from a complex distribution and is not predetermined, directly proving this inequality is challenging. We approximate the optimal update matrix $\overline{E}$ through the update matrix of full fine-tuning (FFT), denoted by $\widetilde{E}$, such that:

$$\overline{E} \approx \widetilde{E} = \Delta W = W_{\text{fft}} - W_0,$$

where $W_{\text{fft}}$ denotes the weights of the fully fine-tuned models.

Directly comparing the singular values of $\widetilde{E} \oslash W_0$ and $\widetilde{E}$ may be influenced by the norm of $W_0$. To mitigate this, we compute the normalized singular values for comparison. Additionally, we analyze the effective rank as described in Appendix C to provide further insights.

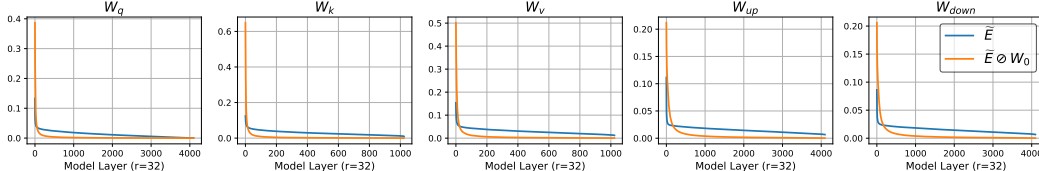

Figure 10: Normalized singular values of layers for $\widetilde{E}$ and $\widetilde{E} \oslash W_0$.

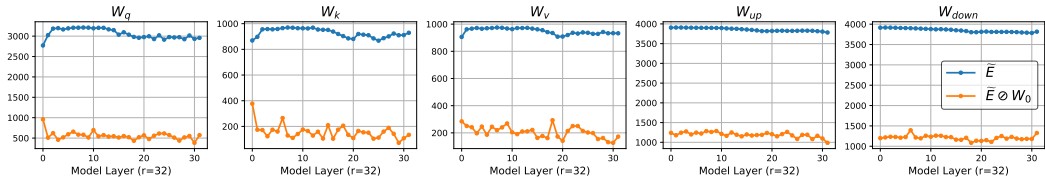

Figure 11: Effective rank across layers for $\widetilde{E}$ and $\widetilde{E} \oslash W_0$.

The singular values of $\widetilde{E} \oslash W_0$ exhibit a faster rate of decline compared to those of $\widetilde{E}$, as demonstrated in Figure 10. This comparison uses normalized singular values to account for the influence of $W_0$'s scaling. Furthermore, Figure 11 demonstrates that the effective rank of $\widetilde{E} \oslash W_0$ is lower than that of $\widetilde{E}$. Those findings indicate that $\widetilde{E} \oslash W_0$ aligns more effectively with low-rank approximations compared to $\widetilde{E}$, supporting the conclusion that HiRA can leverage the Hadamard product to fully utilize the knowledge embedded in the full-rank $W_0$.

