# OpenReview forum: "HiRA: Parameter-Efficient Hadamard High-Rank Adaptation for Large Language Models"
_ICLR.cc/2025/Conference — ICLR 2025 Oral_

### Official Review · Reviewer_m6FX · 2024-11-02

**Soundness:** 4
**Presentation:** 3
**Contribution:** 3
**Rating:** 8
**Confidence:** 4

**Summary:**

This paper introduces Hadamard High-Rank Adaptation (HiRA), a novel parameter-efficient fine-tuning (PEFT) method for Large Language Models (LLMs). HiRA aims to enhance the adaptability of LLMs by using a Hadamard product to retain high-rank update parameters, addressing the limitations of low-rank adaptation methods like LoRA. The authors demonstrate HiRA's effectiveness through extensive experiments and ablation studies, showing improved performance across various tasks compared to LoRA and its variants.

**Strengths:**

1. The paper introduces a new approach to PEFT by applying the Hadamard product to LLM adaptation, which is an original contribution to the field. Compared with LoRA, this method, HiRA, addresses a key limitation of LoRA by enabling high-rank updates, potentially improving the model's capacity to adapt to new tasks.

2. This method maintains a comparable number of trainable parameters to LoRA while enhancing performance, offering a cost-effective solution without additional inference overhead.

3. The paper offers a theoretical basis for the proposed method, and provides experimental results and ablation studies to demonstrate the effectiveness of the method.

**Weaknesses:**

The paper focuses primarily on comparison with LoRA and its variants. It would be beneficial to see comparisons with a broader range of PEFT methods. And it's unclear how well it generalizes across different model architectures or scales.

**Questions:**

1. How does HiRA perform on very large language models (e.g., models with hundreds of billions of parameters)? Are there any scaling challenges?

2. Have you explored the potential of combining HiRA with other PEFT techniques? Could this lead to even better performance?

---

> ### Author Response · Authors · 2024-11-23
> **Response by Authors**
>
> Thank you for your constructive comments. Below we have made responses to your comments. If you have any further comment, please feel free to let us know and we are more than glad to discuss with you.
>
> > Weakness: ... and it's unclear how well it generalizes across different model architectures or scales.
> >
> > Q1: How does HiRA perform on very large language models (e.g., models with hundreds of billions of parameters)? Are there any scaling challenges?
> ### Answer to Q1 and Weakness:
>
> Testing HiRA on models with hundreds of billions of parameters would provide valuable insights into its scalability; however, due to resource constraints, our experiments primarily focus on 7B and 8B models. Moreover, **we extended our evaluation to LLaMA-2-13B**, which is a larger model for assessing HiRA's scalability and performance.
>
> The following table compares HiRA against other PEFT methods, including P-Tuning, Prompt Tuning, MoRA, LoRA, and DoRA, on commonsense reasoning tasks. According to the results, we can see that **HiRA achieves the best average performance.**
>
> | Model       | Method        | Params (\%) | BoolQ     | PIQA      | SIQA      | ARC-c     | ARC-e     | OBQA      | HellaS    | WinoG     | Average   |
> |-------------|---------------|-------------|-----------|-----------|-----------|-----------|-----------|-----------|-----------|-----------|-----------|
> | Llama-2-13B | P-Tuning      | 0.6010      | 62.17     | 0.11      | 0.00      | 0.09      | 0.00      | 0.00      | 0.02      | 0.00      | 7.80      |
> | Llama-2-13B | Prompt Tuning | 0.0003      | 59.20     | 16.70     | 33.01     | 17.49     | 19.82     | 24.60     | 6.31      | 50.36     | 28.44     |
> | Llama-2-13B | MoRA $(r=32)$ | 0.6692      | **75.78** | 83.24     | 80.91     | 76.62     | 88.55     | 83.80     | 29.99     | 85.48     | 75.55     |
> | Llama-2-13B | LoRA $(r=32)$ | 0.6806      | 75.50     | 82.43     | 80.30     | 76.45     | 88.47     | 83.20     | 29.61     | 83.98     | 74.99     |
> | Llama-2-13B | DoRA $(r=32)$ | 0.6806      | 75.35     | 82.92     | **81.88** | 77.05     | 88.51     | 85.00     | 29.91     | 86.35     | 75.87     |
> | Llama-2-13B | HiRA $(r=32)$ | 0.6806      | 71.45     | **86.29** | 80.30     | **77.82** | **90.49** | **85.70** | **89.45** | **86.59** | **83.51** |
>
> ---
> > Weakness: ... It would be beneficial to see comparisons with a broader range of PEFT methods. ...
> >
> > Q2: Have you explored the potential of combining HiRA with other PEFT techniques? Could this lead to even better performance?
> ### Answer to Q2 and Weakness:
> In our submission, we have explored combining HiRA with other PEFT techniques to assess its potential for further performance improvements. Specifically, **in Section 6.6, we evaluated the combination of HiRA and LoRA, referred to as HiLoRA.** As shown in Table 6 and also the following table, HiLoRA improves the average accuracy on commonsense reasoning tasks for LLaMA-3-8B from 86.72 (HiRA) and 80.79 (LoRA) to 87.48. This demonstrates that combining HiRA and LoRA can leverage the strengths of both approaches for enhanced performance.
>
> To further explore such combinations, we also **evaluated the combination of HiRA with Prompt Tuning, which is called HiPT here**. The results in the following table show that HiPT significantly outperforms Prompt Tuning (average accuracy of 80.69 vs. 36.96), but it underperforms HiRA and LoRA. This suggests that while HiRA can complement certain methods, the effectiveness of the combination also depends on the individual strengths of the base approaches in the combination.
>
> | Method                    | Params (\%) | BoolQ     | PIQA      | SIQA      | ARC-c     | ARC-e     | OBQA      | HellaS    | WinoG     | Average   |
> |---------------------------|-------------|-----------|-----------|-----------|-----------|-----------|-----------|-----------|-----------|-----------|
> | Prompt Tuning             | 0.0010      | 56.85     | 45.05     | 36.13     | 31.57     | 32.74     | 29.20     | 14.01     | 50.12     | 36.96     |
> | LoRA $(r=32)$             | 0.7002      | 70.80     | 85.20     | 79.90     | 71.20     | 84.20     | 79.00     | 91.70     | 84.30     | 80.79     |
> | HiRA $(r=32)$             | 0.7002      | 75.40     | 89.70     | 81.15     | 82.90     | 93.27     | **88.32** | 95.36     | 87.70     | 86.72     |
> | HiLoRA $(r_1=20, r_2=12)$ | 0.7002      | **75.84** | **90.42** | **82.14** | **84.22** | **93.73** | 87.80     | **96.45** | **89.27** | **87.48** |
> | HiPT$(r=32)$              | 0.7006      | 69.42     | 82.92     | 77.53     | 74.91     | 86.28     | 77.60     | 90.62     | 86.27     | 80.69     |

---

> > ### Comment · Reviewer_m6FX · 2024-11-26
> > **thanks for detailed response**
> >
> > my questions were addressed, no further questions.

---

> ### Author Response · Authors · 2024-11-25
>
> We deeply appreciate the time and effort you have dedicated to reviewing our manuscript. As the rebuttal period is coming to an end, we would like to kindly remind you that we have submitted responses to your comments. We would be most grateful if you could confirm whether our responses have sufficiently addressed your concerns. If you have any further feedback or additional concerns, please do not hesitate to let us know.

---

### Official Review · Reviewer_q3zD · 2024-11-03

**Soundness:** 3
**Presentation:** 3
**Contribution:** 3
**Rating:** 8
**Confidence:** 4

**Summary:**

This paper proposes the use of the Hadamard product in the new technique HiRA to parametrize _high-rank_ updates to the base model, based on LoRA, while using the same parameter count. Empirically HiRA outperforms LoRA on various tasks. The authors also provided analyses showing 1) the updates in HiRA are indeed high rank and 2) prominent singular values of HiRA trend more like full fine-tuning than LoRA. Finally the authors look into combining LoRA and HiRA as HiLoRA.

**Strengths:**

Clear presentation that empathizes the simple insight that the Hadamard product yields high-rank matrices. Extensive experiments. Convincing ablation studies.

**Weaknesses:**

More theoretical analysis would be useful. For example, more clarification on why Theorem 1 is important will help readers a lot.

**Questions:**

Did the authors consider alternative parametrization of the right operand of the Hadamard product? For example instead of W_0 maybe 'smooth' it with adjacent layers / other attention components within the same layer?

---

> ### Author Response · Authors · 2024-11-23
> **Response by Authors**
>
> Thank you for your constructive comments. Below we have made responses to your comments. If you have any further comment, please feel free to let us know and we are more than glad to discuss with you.
>
> > Question: Did the authors consider alternative parametrization of the right operand of the Hadamard product? For example instead of W_0 maybe 'smooth' it with adjacent layers / other attention components within the same layer?
>
> ### Answer to Question:
>
> According to your suggestion, we experimented with a **smoothed version of $W_0$ by incorporating information from adjacent layers**. Specifically, the smoothed $W_0$ ($R=W_{smooth}$) is computed as the **average of parameters in the upper, current, and lower layers** for the same transformer component. The results, summarized in the table below, show that the smoothed $W_0$ achieves an average score of 83.71, slightly lower than the original $W_0$, which achieves 86.72. While such smoothing provides competitive performance, the original $W_0$ retains a clear advantage, likely due to its direct alignment with the pretrained weights, preserving task-specific expressiveness and parameter efficiency.
>
>
> | Model          | BoolQ     | PIQA      | SIQA      | ARC-c     | ARC-e     | OBQA      | HellaS    | WinoG     | Average   |
> |----------------|-----------|-----------|-----------|-----------|-----------|-----------|-----------|-----------|-----------|
> | $R=W_{smooth}$ | 72.02     | 83.84     | 79.84     | 81.23     | 90.24     | 85.60     | 92.07     | 84.85     | 83.71     |
> | $R=W_0$        | **75.40** | **89.70** | **81.15** | **82.90** | **93.27** | **88.32** | **95.36** | **87.70** | **86.72** |
>
> Moreover, we conducted **an additional experiment by reducing the rank of $R$** on LLaMA-3-8B across commonsense reasoning tasks. Specifically, $R$ is derived from $W_0$, the pretrained weight matrix, using SVD decomposition. We retained only the top-$v$ singular values of $U$, $S$, and $V$ to form a reduced-rank version, $W_{0,{[:v]}}$. The results, presented in the following table, show that increasing the rank of $R$ generally improves the performance across tasks.
>
>
> | Model              | $Rank(R)$ | BoolQ     | PIQA      | SIQA      | ARC-c     | ARC-e     | OBQA      | HellaS    | WinoG     | Average   |
> |--------------------|-----------|-----------|-----------|-----------|-----------|-----------|-----------|-----------|-----------|-----------|
> | $R=W_{0,{[:32]}}$  | 32        | 73.67     | 88.63     | 80.04     | 80.97     | 93.18     | 86.00     | 94.31     | 86.58     | 85.42     |
> | $R=W_{0,{[:128]}}$ | 128       | 74.16     | 89.23     | **82.04** | 80.89     | 93.10     | 87.20     | 95.11     | 87.45     | 86.15     |
> | $R=W_{0,{[:512]}}$ | 512       | 74.34     | **89.88** | 81.99     | 82.25     | **93.86** | 88.20     | **95.38** | 87.61     | 86.69     |
> | $R=W_0$            | 2852      | **75.40** | 89.70     | 81.15     | **82.90** | 93.27     | **88.32** | 95.36     | **87.70** | **86.72** |
>
> In summary, all those experiments show that several alternatives of $R$ could possess good performance, and among them, our choice, $R=W_0$, have better average performance, showing its competitiveness.
>
> ---
>
> > Weakness: More theoretical analysis would be useful. For example, more clarification on why Theorem 1 is important will help readers a lot.
> ### Answer to Weakness:
>
> Theorem 1 is crucial because it provides theoretical insight into the expressive power of HiRA by establishing an upper bound on its approximation error. Although the weighted low-rank approximation problem is NP-Hard and requires certain relaxations in our proof, this do not diminish the significance of Theorem 1.
> - Specifically, unlike LoRA, our derived upper bound is proportional to $\sigma_{r+1}(\overline{E} \oslash W_0)$. This proportionality indicates that when the optimal update matrix for a downstream task is related to the pre-trained weights $W_0$, **HiRA can effectively leverage the Hadamard product to directly utilize the knowledge embedded in the full-rank $W_0$**, facilitating rapid adaptation to new task. In contrast, methods like LoRA do not directly exploit this relationship.
> - Additionally, our experimental results, including random initialization of $R$ (Section 6.3) and the additional experiments with alternative $R$ (e.g., 'smooth' $W_0$), further validate the theoretical insights. These findings explain why HiRA can achieve superior performance by capturing essential information from $W_0$ during fine-tuning.

---

> ### Author Response · Authors · 2024-11-25
>
> We deeply appreciate the time and effort you have dedicated to reviewing our manuscript. As the rebuttal period is coming to an end, we would like to kindly remind you that we have submitted responses to your comments. We would be most grateful if you could confirm whether our responses have sufficiently addressed your concerns. If you have any further feedback or additional concerns, please do not hesitate to let us know.

---

> > ### Comment · Reviewer_q3zD · 2024-11-26
> > **Thanks for the response!**
> >
> > And thanks for the additional experiments. However more theoretical justification is needed to show that Thm 1 is important.
> >
> > For example, can the authors show that $\sigma_{r+1}(\overline{E} \oslash W_0) < \sigma_{r+1}(\overline{E})$ for a typical $W_0$?

---

> ### Author Response · Authors · 2024-11-27
>
> Thank you for your insightful feedback. We greatly appreciate the opportunity to provide a more thorough theoretical justification for the importance of Theorem 1. If you have any further questions or need additional clarification, please do not hesitate to reach out.
>
> > Question: More theoretical justification is needed to show that Thm 1 is important. For example, can the authors show that $\sigma_{r+1}(\overline{E} \oslash W_0)< \sigma_{r+1}(\overline{E})$ for a typical $W_0$?
>
> ### Answer to Question:
>
> The Hadamard (element-wise) division $\overline{E} \oslash W_0$ normalizes the entries of $\overline{E}$ by the corresponding entries in $W_0$. Since the optimal update matrix $\overline{E}$ comes from a complex distribution and is not predetermined, directly proving this inequality is challenging. However, we provide two ways to approxiamate it to substantiate this claim based on empirical evidence.
> 1. Our experiments demonstrate that HiRA produces a high-rank update matrix by elementwisely multiplying a low-rank matrix with $W_0$. This is evident from:
>     - **Number of Significant Singular Values**: As shown in Figure 4, the update matrix exhibits multiple large singular values, indicating its high rank.
>     - **Effective Rank Analysis**: The newly added Figure 8 confirms that HiRA captures high-rank structures.
>
>     **If we approximate the optimal update matrix $\overline{E}$ by using that (denoted by $\widetilde{E}$) in HiRA**, we obviously observe $\sigma_{r+1}(\widetilde{E} \oslash W_0)=\sigma_{r+1}(W_{hi})=0$ and $\sigma_{r+1}(\widetilde{E})=\sigma_{r+1}(W_0 \odot W_{hi}) > 0$, hence, the inequailty holds.
>
> 2. **The optimal update matrix $\overline{E}$ can also be approximated based on FFT**:
>     $$
>     \overline{E} \approx \widetilde{E} = \Delta W = W_{\text{fft}}-W_{0},
>     $$
>     where $W_{\text{fft}}$ denotes the weights of the fully fine-tuned model. To compare $\widetilde{E}$ and $\widetilde{E} \oslash W_0$, we analyze:
>     - **Rate of Singular Value Decline**: Singular values of $\widetilde{E} \oslash W_0$ decrease faster than those of $\widetilde{E}$, as shown in the newly added Figure 10 in Appendix F, using normalized singular values to account for the scaling of $W_0$.
>     - **Effective Rank Comparison**: Newly added Figure 11 in Appendix F illustrates that the effective rank of $\widetilde{E} \oslash W_0$ is smaller than that of $\widetilde{E}$.
>
>     Those findings suggest that $\widetilde{E} \oslash W_0$ aligns better with low-rank approximations than $\widetilde{E}$, supporting the conclusion that “HiRA can effectively leverage the Hadamard product to directly utilize the knowledge embedded in the full-rank $W_0$.”

---

### Official Review · Reviewer_P2fT · 2024-11-05

**Soundness:** 3
**Presentation:** 3
**Contribution:** 3
**Rating:** 8
**Confidence:** 4

**Summary:**

The paper proposes Hadamard High-Rank Adaptation (HiRA), a parameter-efficient fine-tuning (PEFT) method for LLMs, to address limitations of Low-Rank Adaptation (LoRA) for complex tasks. The authors argue that while LoRA's expressiveness can be limited by its low-rank nature, which may restrict the model's ability to adapt to complex tasks. HiRA addresses this limitation by applying a Hadamard product between the low-rank update matrices and the frozen pretrained weights. The authors claim that this method effectively creates high-rank updates and enhance the expressiveness of the LLMs without significantly increasing the number of trainable parameters.

**Strengths:**

The paper introduces a creative way of using the Hadamard product to achieve high-rank adaptation within PEFT. This approach is an important contribution to address the common expressiveness limitation of LoRA-like PEFT methods. The authors show HiRA's consistent improvements over LoRA and other baselines which demonstrate that HiRA not only theoretically sounds but also practically effective. The ablation studies, such as singular value distribution analysis, show how HiRA's parameter dynamics behaves somewhat similarly to those of full fine-tuning and provide insights into the understanding of HiRA's expressiveness. Furthermore, the paper also compares HiRA not only to LoRA but also to another high-rank adaptation method called MoRA to justify the claim that its Hadamard product approach has unique advantages in achieving high-rank adaptability. Finally, HiRA seems to be lightweight and computationally efficient, which is attractive for practical applications.

**Weaknesses:**

First, my major concern is the connection between high rank, expressiveness and improved performance. Although the paper attempts to address this connection through ablation studies and performance improvements in experiments, it lacks a rigorous explanation that directly links high rank to expressiveness and explains why this is essential for improved performance in complex tasks. In other words, without showing how the observed rank impacts various task complexities, it's unclear if performance gains are solely due to the high rank, not other factors. For example, it's possible that the similarity in singular values of HiRA compared to FFT is due to an artifact of the data or tasks rather than an inherent property of high-rank adaptations.

Second, the paper lacks an in-depth analysis to show unique advantages of HiRA's high-rank nature. In particular, it's unclear if the improved performance of HiRA over MoRA is due solely to high-rank structure of Hadamard product, not other aspects of HiRA. Without detailed comparison of HiRA with other high-rank approaches such as MoRA, for example, showing distinctive singular value patterns of HiRA compared to those of MoRA, it's not easy to see unique benefits of HiRA's high-rank structure.

Finally, the paper provides limited insights into HiRA's generalization ability. In particular, while a high-rank structure might improve adaptability, it could also be potentially overfitting. The paper would benefit from evaluation of HiRA on, say, tasks requiring transfer learning.

**Questions:**

1. Is there a way to gain insights that using HiRA's Hadamard product is a better choice than other high-rank methods.
2. Can the authors justify that the singular value patterns of HiRA closely correlated to expressiveness properties that benefit tasks? Perhaps a detailed comparison of HiRA and MoRA in this context might help.
3. Are there attributes of HiRA other than high-rank strucutre that may contribute to its improved performance? Again, an in-depth comparison of HiRA and MoRA might help clarify this question.
4. Is HiRA prone to overfitting? Would evaluating HiRA on, say, transfer learning tasks help address this question?
5. Could you provide quantitative evidence of HiRA’s computational efficiency, such as memory usage, training time, and inference latency?

---

> ### Author Response · Authors · 2024-11-23
> **Response by Authors, Part 1**
>
> Thank you for your constructive comments. Below we have made responses to your comments. If you have any further comment, please feel free to let us know and we are more than glad to discuss with you.
>
> > Weaknesses:
> > First, my major concern is the connection between high rank, expressiveness and improved performance. Although the paper attempts to address this connection through ablation studies and performance improvements in experiments, it lacks a rigorous explanation that directly links high rank to expressiveness and explains why this is essential for improved performance in complex tasks. In other words, without showing how the observed rank impacts various task complexities, it's unclear if performance gains are solely due to the high rank, not other factors. For example, it's possible that the similarity in singular values of HiRA compared to FFT is due to an artifact of the data or tasks rather than an inherent property of high-rank adaptations.
> >
> > Second, the paper lacks an in-depth analysis to show unique advantages of HiRA's high-rank nature. In particular, it's unclear if the improved performance of HiRA over MoRA is due solely to high-rank structure of Hadamard product, not other aspects of HiRA. Without detailed comparison of HiRA with other high-rank approaches such as MoRA, for example, showing distinctive singular value patterns of HiRA compared to those of MoRA, it's not easy to see unique benefits of HiRA's high-rank structure.
> >
> > Finally, the paper provides limited insights into HiRA's generalization ability. In particular, while a high-rank structure might improve adaptability, it could also be potentially overfitting. The paper would benefit from evaluation of HiRA on, say, tasks requiring transfer learning.
>
> ### Answer to Weaknesses:
>
> We sincerely appreciate your thorough review. You have highlighted three primary weaknesses, each corresponding to specific questions. Specifically, **the first weakness relates to Q1, Q2, and Q3**, **while the remaining two weaknesses correspond to Q4 and Q5**, respectively. Below, we provide a detailed response addressing each of your questions point by point, which could be responses to weaknesses.
>
> ---
>
> > Q1: Is there a way to gain insights that using HiRA's Hadamard product is a better choice than other high-rank methods.
>
> ### Answer to Q1:
>
>
> To provide such insights, our analysis focuses on comparing the fine-tuning dynamics of HiRA with other high-rank methods in relation to full fine-tuning (FFT), which, while highly accurate, is computationally intensive. In Figures 4 and 5 of the updated version of the manuscript, we supplemented our comparisons with some high-rank PEFT methods, such as **LoRA with ranks 128 and 512** and **MoRA**. From those results, we can have the following observations:
> - For LoRA with increasing ranks, the number of significant singular values (greater than 0.005) remains limited. This implies that merely increasing the rank of LoRA does not align LoRA's fine-tuning dynamics with those of FFT. For LoRA with rank=512, the Frobenius norm decreases to a level similar to FFT, indicating that the magnitude of parameter updates is relatively low, thereby preserving the robustness of the pre-trained model.
> - For MoRA, this method effectively produces an almost full-rank update matrix with relatively large singular values. The high Frobenius norm associated with MoRA indicates substantial parameter updates, which may adversely affect the robustness of the pre-trained model by **deviating too much from the original parameters**.
> - For HiRA with the Hadamard product, the singular value distribution closely mirrors that of FFT, suggesting that HiRA captures similar fine-tuning dynamics to those achieved by full fine-tuning.

---

> ### Author Response · Authors · 2024-11-23
> **Response by Authors, Part 2**
>
> > Q2: Can the authors justify that the singular value patterns of HiRA closely correlated to expressiveness properties that benefit tasks? Perhaps a detailed comparison of HiRA and MoRA in this context might help.
> ### Answer to Q2:
>
> Thank you for your valuable suggestion. We have added a detailed comparison of HiRA and MoRA in terms of singular value patterns, as discussed in our response to Q1. Additionally, similar to [r1], we conducted further experiments to analyze the effective rank [r2], which is defined in the next paragraph, of different methods, as presented in Figure 8 of Appendix C.
>
> Compared to simply calculating the rank or counting the number of singular values above a given threshold, the **effective rank accounts for the full singular value spectrum**. For a matrix $A$ of size $M \times N$ with $K=\min\{M,N\}$, the effective rank is defined as follows:
>     $$
>     \operatorname{erank}(A)=\exp \left(-\sum_{k=1}^K p_k \log p_k \right), \quad p_k = \frac{\sigma_k}{\|\sigma\|_1},
>     $$
> where $\sigma_k$ denotes the $k$-th largest singular value, $\sigma=(\sigma_1,\ldots,\sigma_K)^T$, and $\|\cdot \|_1$ is the ${l}_1$-norm of a vector. It is easy to see that a high effective rank indicates that singular values are close to each other.
>
> From the results, we can observe that **the update matrix obtained by FFT achieves the highest effective rank**. For the query component, MoRA shows behavior closer to FFT, while for the value component—and especially for the projection layer (up and down components)—**HiRA exhibits better**. Conversely, even LoRA with rank=512 offers a relatively lower effective rank.
>
> In summary, MoRA and HiRA behave differently in terms of the effective rank as well as the distributions of singular values, and HiRA behaves more close to full fine-tuning than other high-rank PEFT methods, which could be one reason that HiRA earns better performance among high-rank PEFT methods.
>
>
> [r1] Shuttleworth, Reece, et al. LoRA vs Full Fine-tuning: An Illusion of Equivalence, arXiv:2410.21228.
>
> [r2] Roy O, Vetterli M. The effective rank: A measure of effective dimensionality, European Signal Processing Conference, 2007.

---

> ### Author Response · Authors · 2024-11-23
> **Response by Authors, Part 3**
>
> ---
> > Q3: Are there attributes of HiRA other than high-rank strucutre that may contribute to its improved performance? Again, an in-depth comparison of HiRA and MoRA might help clarify this question.
> ### Answer to Q3:
>
> HiRA’s improved performance can be attributed to several unique attributes beyond its high-rank structure:
>
> 1. As discussed in our responses to Q1 and Q2, HiRA differs from MoRA and high-rank LoRA in the following terms:
>    - The number of large singular values in the update parameter matrix.
>    - The Frobenius norm of the update parameter matrix.
>    - The effective rank of the update parameter matrix.
>
>    Those factors contribute to HiRA demonstrating characteristics more akin to FFT, enhancing its performance.
>
> 2. In Eq. (3) of our manuscript, we introduce a general formulation for HiRA based on the matrix $R$ and then we set $R$ to be the pre-trained weight matrix $W_0$. By doing so, HiRA **directly utilizes the knowledge embedded in the pre-trained weights** via the Hadamard product. This allows HiRA to adapt quickly to downstream tasks by modulating existing weights rather than introducing entirely new parameters.
>
>     To further validate our approach, we conducted ablation experiments exploring alternative configurations for $R$:
>     - Random matrix (Section 6.3).
>     - Low-rank approximation of $W_0$ (**newly added**, Appendix E.1 in the updated manuscript).
>     - The average of pre-trained parameters in adjacent layers (**newly added**, Appendix E.2 in the updated manuscript).
>
>     We summarize the results in the following table, where $W_{0,{[:v]}}$ denotes a rank-$v$ approximation of $W_0$ based on top-$v$ singular values and the corresponding singular vectors of $W_0$ and $W_{smooth}$ is the average of parameters from the upper, current, and lower layers of the same transformer component. Hence, $R=W_{0,{[:v]}}$ corresponds to the second alternative mentioned above and $R=W_{smooth}$ is for the third alternative mentioned above. According to the results, we can see that setting $R$ to be $W_0$ is a simple yet effective choice.
>
> | Model              | $Rank(R)$ | BoolQ     | PIQA      | SIQA      | ARC-c     | ARC-e     | OBQA      | HellaS    | WinoG     | Average   |
> |--------------------|-----------|-----------|-----------|-----------|-----------|-----------|-----------|-----------|-----------|-----------|
> | $R=W_{smooth}$     | 2863      | 72.02     | 83.84     | 79.84     | 81.23     | 90.24     | 85.60     | 92.07     | 84.85     | 83.71     |
> | $R=W_{0,{[:32]}}$  | 32        | 73.67     | 88.63     | 80.04     | 80.97     | 93.18     | 86.00     | 94.31     | 86.58     | 85.42     |
> | $R=W_{0,{[:128]}}$ | 128       | 74.16     | 89.23     | **82.04** | 80.89     | 93.10     | 87.20     | 95.11     | 87.45     | 86.15     |
> | $R=W_{0,{[:512]}}$ | 512       | 74.34     | **89.88** | 81.99     | 82.25     | **93.86** | 88.20     | **95.38** | 87.61     | 86.69     |
> | $R=W_0$            | 2852      | **75.40** | 89.70     | 81.15     | **82.90** | 93.27     | **88.32** | 95.36     | **87.70** | **86.72** |
>
> ---
> > Q4: Is HiRA prone to overfitting? Would evaluating HiRA on, say, transfer learning tasks help address this question?
> ### Answer to Q4:
> To address concerns about overfitting, we trained HiRA and LoRA separately on eight commonsense reasoning tasks and evaluated their performance on all the eight tasks.
>
> As shown in Figure 9 of Appendix D in the updated manuscript, HiRA demonstrates strong transferability across tasks, achieving high scores on related tasks. This indicates its ability to generalize effectively beyond the specific task it was trained on.
>
> In comparison, LoRA exhibits more pronounced overfitting. While it performs well on its training task (e.g., boolq-lora scores 71.44 on boolq), its performance drops significantly on other tasks. In contrast, HiRA maintains respectable transferability, as seen with arce-hira, which achieves 64.23 on siqa, 70.20 on obqa, and 58.56 on winog, outperforming LoRA in these cases.
>
> HiRA also excels at harder tasks requiring more complex reasoning, such as HellaSwag (hellas) and WinoGrande (winog). On the other hand, LoRA performs relatively better on simpler tasks like BoolQ and PIQA, but its overfitting tendency causes significant drops in performance on unrelated tasks.
>
> In summary, HiRA shows superior generalization and is less prone to overfitting compared to LoRA.

---

> ### Author Response · Authors · 2024-11-23
> **Response by Authors, Part 4**
>
> > Q5: Could you provide quantitative evidence of HiRA’s computational efficiency, such as memory usage, training time, and inference latency?
> ### Answer to Q5:
> Quantitative evidence of HiRA's computational efficiency, including **training time and memory usage**, have been provided in **Appendix A.6 and Table 12**. Those results show that HiRA and LoRA have similar training time and memory requirements. For inference latency, **HiRA can be seamlessly merged into the original LLM**, resulting in **no additional inference overhead** compared to the original LLM.

---

> ### Author Response · Authors · 2024-11-25
>
> We deeply appreciate the time and effort you have dedicated to reviewing our manuscript. As the rebuttal period is coming to an end, we would like to kindly remind you that we have submitted responses to your comments. We would be most grateful if you could confirm whether our responses have sufficiently addressed your concerns. If you have any further feedback or additional concerns, please do not hesitate to let us know.

---

> > ### Comment · Reviewer_P2fT · 2024-11-28
> > **Thanks for the detailed response!**
> >
> > My concerns were addressed. I have no further question.

---

### Public Comment · ~Yao_Ni1 · 2025-03-13
**Existing works on Hadamard Product-based LoRA are not discussed.**

Dear Authors,

Congratulations on having your paper accepted! Your work is very interesting.

I noticed that the form of the Hadamard product between LoRA weights and pretrained weight presented in this work has already been explored in existing methods (Fig. 1 in [1] and LoRA$_\text{mul}$ in [2]). While this work introduces the Hadamard product form from a new perspective, properly citing and acknowledging these prior works would be appropriate and help provide a more comprehensive context for readers.

Thank you for your consideration!

Best regards,

Yao

[1] Wen, Yeming, and Swarat Chaudhuri. "Batched Low-Rank Adaptation of Foundation Models." The International Conference on Learning Representations 2024.

[2] Ni, Yao, Shan Zhang, and Piotr Koniusz. "PACE: Marrying Generalization in PArameter-efficient Fine-Tuning with Consistency rEgularization." Advances in Neural Information Processing Systems 37 (2024): 61238-61266.

---

> ### Public Comment · ~Qiushi_Huang1 · 2025-04-01
>
> Dear Yao,
>
> Thank you for your thoughtful feedback and for pointing out these relevant prior works. We truly appreciate your insights and agree that properly acknowledging these contributions helps provide a more comprehensive context for our study.
>
> Following your suggestion, we have revised the Related Work section to explicitly cite FLoRA and PACE, recognizing their exploration of element-wise multiplication in adapters and its advantages over classic LoRA. However, those works, along with ours, focus on different objectives: cost efficiency, generalization, and expressive power, which makes the three works different.
>
> We sincerely appreciate your constructive input, which has helped us refine our paper and improve its clarity. Thank you for taking the time to engage in this discussion.
>
> Best regards,
>
> Authors of HiRA

---

### Meta-Review · Area_Chair_4si1 · 2024-12-12

**Metareview:**

This paper introduces HiRA (Hadamard High-Rank Adaptation), a parameter-efficient fine-tuning method that improves the adaptability of large language models (LLMs). HiRA utilizes the Hadamard product to preserve high-rank update parameters, achieving greater expressiveness than LoRA. Empirical results demonstrate that HiRA consistently outperforms LoRA and its variants across various tasks, supported by extensive ablation studies. The paper is clearly written and easy to follow, with detailed experiments. The authors have thoroughly addressed reviewers' concerns and questions. Therefore, I recommend acceptance.

**Additional Comments On Reviewer Discussion:**

The authors have addressed the reviewers' concerns effectively. During the rebuttal process, they made several key revisions to address the feedback, including:
- Expanded comparisons
- Ablation studies
- Generalization and overfitting analysis
- Efficiency and scalability improvements
- Integration with other PEFT techniques
- Enhanced theoretical justification

Overall, this is a well-executed and commendable piece of work.

---

### Decision · Program_Chairs · 2025-01-22

Accept (Oral)